# Oleuropein Ameliorates Bleomycin-Induced Pulmonary Fibrosis in Mice by Targeting TGF-β1 Signaling Pathway

**DOI:** 10.3390/biom15091211

**Published:** 2025-08-22

**Authors:** Liang Zhang, Zhigang Liu, Yayue Hu, Xueze Liu, Zhongyi Yang, Yuming Liu, Ran Jiao, Xiaoting Gu, Weidong Zhang, Xiaohe Li, Honggang Zhou

**Affiliations:** 1Department of Thoracic Surgery, Tianjin First Central Hospital, College of Pharmacy, Nankai University, Tianjin 300353, China; zhangliang001@tmu.edu.cn (L.Z.); 1120220741@mail.nankai.edu.cn (Z.L.); 1120240971@mail.nankai.edu.cn (Y.H.); 2120241773@mail.nankai.edu.cn (X.L.); 2120231679@mail.nankai.edu.cn (Z.Y.); 2120231738@mail.nankai.edu.cn (Y.L.); 2120231714@mail.nankai.edu.cn (R.J.); guxiaoting@nankai.edu.cn (X.G.); honggang.zhou@nankai.edu.cn (H.Z.); 2Tianjin Key Laboratory of Molecular Drug Research, Tianjin International Joint Academy of Biomedicine, Tianjin 300457, China

**Keywords:** idiopathic pulmonary fibrosis, oleuropein, TGF-β1, fibroblasts, FAP-α

## Abstract

Idiopathic pulmonary fibrosis (IPF) is a chronic, progressive interstitial lung disease characterized by the accumulation of fibrotic tissue in the lungs, leading to impaired gas exchange and respiratory failure, with a poor prognosis and limited treatment options. Oleuropein, a compound extracted from olive leaves, demonstrates a range of pharmacological activities, including benefits for non-alcoholic fatty liver disease and cardiac fibrosis. This study investigates the therapeutic potential of oleuropein for IPF and its underlying mechanisms. We first established a bleomycin-induced mouse model of pulmonary fibrosis and evaluated the in vivo efficacy of oleuropein. Our findings demonstrated that oleuropein significantly alleviated lung fibrosis and improved pulmonary function. Through in vitro experiments, we found that oleuropein inhibited TGF-β1-induced fibroblast migration, activation, autophagy, and apoptotic resistance, and mechanistically, oleuropein could regulate the TGF-β1/Smad and TGF-β1/mTOR signaling pathways in fibroblasts. Additionally, molecular docking analysis indicated that FAP-α is a potential target of oleuropein, displaying strong binding affinity. The effects of oleuropein on fibroblasts were markedly disrupted in FAP-α knockout cells. In conclusion, oleuropein exerts its beneficial effects by targeting FAP-α and inhibiting TGF-β1-related signaling pathways, improving the pathological characteristics of pulmonary fibrosis in mouse models, and demonstrating promising application prospects for the treatment of IPF.

## 1. Introduction

Idiopathic pulmonary fibrosis (IPF) is a chronic fibrotic interstitial pneumonia of unknown etiology, characterized by the radiological and histological features of usual interstitial pneumonia (UIP). IPF typically occurs in older individuals and is marked by progressive dyspnea and declining lung function, leading to a poor prognosis [1]. The median survival time for patients with IPF is approximately 1 to 2 years [2,3,4]. The global incidence of IPF is estimated to be between one to thirteen cases per 100,000 people, with a prevalence of three to forty-five cases per 100,000 [5]. As the global population continues to age, increased awareness of IPF’s clinical presentation, along with the emergence of various risk factors, is expected to contribute to a rising global burden of the disease. The pathogenesis of IPF is thought to result from a complex interplay of genetic and environmental factors, as well as comorbid conditions [6]. Viral infections (such as hepatitis C virus, adenovirus, SARS-CoV-2, and herpesvirus) are also significant risk factors that may lead to lung fibrosis, acute exacerbation, and disease progression. Furthermore, comorbidities like gastroesophageal reflux disease, diabetes, and obstructive sleep apnea can also increase the risk of developing IPF [6,7]. The disease course and progression of IPF are heterogeneous and unpredictable. Currently, two medications, pirfenidone and nintedanib, can slow the decline in lung function in IPF patients, but they do not halt disease progression or extend survival.

The pathogenesis of IPF is complex and involves interactions among various cells and signaling pathways. The initial stage of IPF usually involves abnormal repair of alveolar epithelial cells, which promotes inflammation and fibrosis. Overactivation of fibroblasts is also a key factor, as these cells play an important role in normal repair of lung tissue, while in IPF, they lead to abnormal deposition of collagen, which promotes the development of fibrosis. Functional abnormalities in fibroblasts are also closely linked to their resistance to apoptosis and reduced autophagy. Under normal circumstances, apoptosis is a key mechanism for maintaining tissue homeostasis; however, in IPF, fibroblasts exhibit resistance to apoptotic signals. This apoptotic resistance allows fibroblasts to persist and remain active in pathological conditions, further exacerbating the fibrotic process. Additionally, autophagy is an essential mechanism for the clearance of damaged components and maintenance of the energy balance. Its reduction in fibroblasts leads to the accumulation of harmful substances within the cells, impairing normal cellular functions and accelerating the onset of fibrosis.

On the other hand, the aberrant activation of profibrotic factors, such as the TGF-β pathway, also plays a significant role in IPF. This pathway activates downstream Smad and non-Smad signaling pathways, including the MAPK and mTOR pathways. These pathways not only promote the proliferation and migration of fibroblasts but also further inhibit autophagy and enhance cellular resistance to apoptosis. Therefore, a deeper understanding of the functional changes in fibroblasts and the dysregulation of their associated signaling pathways will provide important insights into the pathological mechanisms of IPF and potential therapeutic targets [8].

Oleuropein is a secoiridoid glycoside compound found in plants such as olive leaves, with the potential to prevent various age-related diseases, including those associated with steatohepatitis and fibrosis. Studies have shown that oleuropein can treat non-alcoholic steatohepatitis (NASH) and its progression to fibrosis, significantly reducing liver damage, pathological obesity, and lipid accumulation, while also alleviating hepatitis and fibrosis [9]. In diabetic models, oleuropein has demonstrated the ability to improve cardiac fibrosis and hypertrophy, suppressing the expression of fibrosis-related genes [10]. Additionally, research has indicated that oleuropein has protective effects on chronic obstructive pulmonary disease (COPD) and acute lung injury, primarily exerting its effects through anti-inflammatory and antioxidant mechanisms [11,12]. Industrial extracts of olive leaves have been shown to ameliorate bleomycin-induced pulmonary fibrosis in rats [13]. Therefore, oleuropein plays a significant pharmacological role in the prevention and treatment of fibrosis-related diseases.

Given the central role of fibroblast overactivation in the pathogenesis of pulmonary fibrosis, the activation marker α-SMA has been utilized for the early screening of drugs with antifibrotic effects. Fibroblast activation was induced using TGF-β1, and the impact of compounds from a library of traditional Chinese medicine monomers on α-SMA expression was assessed. The screening results indicated that oleuropein dose-dependently inhibited TGF-β1-induced α-SMA expression. Furthermore, oleuropein demonstrated promising antifibrotic effects on animal models of pulmonary fibrosis. Therefore, this study aims to investigate the therapeutic effects and mechanisms of oleuropein in pulmonary fibrosis, providing new insights for the treatment of this condition.

## 2. Materials and Methods

### 2.1. Mice and Bleomycin Induction of Pulmonary Fibrosis

C57BL/6J male mice, aged 6 to 8 weeks, were purchased from Beijing Vital River Laboratory Animal Technology Co., Ltd. (Shenzhen, China). (Production License Number: SCXK (Jing) 2021-0006). These mice were allowed free access to water and food. The mice were randomly divided into the following groups: NaCl group, bleomycin (BLM) group, nintedanib 60 mg/kg group, Ole 25 mg/kg group, Ole 50 mg/kg group, and Ole 100 mg/kg group, with 5 mice in each group. After acclimatization to the breeding environment, modeling was conducted. Following anesthesia, the mice were fixed and the area disinfected with alcohol, after which an incision was made in the neck using a surgical scalpel. The muscle tissue was separated with forceps to expose the trachea, and a syringe was inserted into the trachea toward the ribs to administer bleomycin (970592, Nippon Kayaku Co., Ltd., Tokyo, Japan) at a dosage of 2 U/kg. After the injection, the mice were held upright and gently tapped on the back to facilitate the distribution of bleomycin in the lungs. The sham surgery group received an injection of saline into the trachea.

Oleuropein is a monoterpenoid with the molecular formula C_25_H_32_O_13_, and was purchased from Chengdu Push Bio-technology Co., Ltd., Chengdu, China, with a purity of >98.0%.

### 2.2. Experimental Design

Thirty mice were included in the experiment. Using a random number table, the mice were divided into the control group, the model group, and the drug group based on their body weight. The entire grouping process was completely random. On the seventh day, we measured the degree of lesions in the mice after they received BLM stimulation using microCT and administered the drug. We chose the marketed drug nintedanib for idiopathic pulmonary fibrosis as the positive drug to measure the stability of the entire system.

### 2.3. Pulmonary Function Testing

After anesthetizing and securing the mice, the area was disinfected with alcohol, and an incision was made in the neck using a surgical scalpel. Muscle tissue was carefully dissected with forceps to expose the trachea. A surgical suture was placed beneath the trachea, and a long needle with a tracheal cannula was inserted. The needle tip carefully punctured the trachea, and the cannula was slowly twisted into place. A surgical suture was used to secure the cannula to the trachea. The tracheal cannula was then connected to a plethysmograph (AniRes2005, Beijing BoiLab Co., Ltd., Beijing, China) to measure pulmonary function data while the mice were in a stable breathing state. The relevant operations were carried out according to the steps provided in the manual of the lung function instrument.

### 2.4. Histological and Immunohistochemical Analyses

During lung tissue collection, the right lung was ligated with surgical sutures, and 4% paraformaldehyde was infused through the trachea to inflate the left lung. The left lung, along with the trachea, were then excised and immersed in 4% paraformaldehyde for 48 h for fixation. Subsequent procedures included dehydration and paraffin embedding. Paraffin sections with a thickness of 4 μm were prepared and placed in an oven at 60 °C for 2 h to ensure it can be used to start dyeing.

The HE staining procedure was conducted according to the manufacturer’s instructions (G1120, Beijing Solarbio Science & Technology Co., Ltd., Beijing, China). The Masson staining procedure was conducted according to the manufacturer’s instructions (G1340, Beijing Solarbio Science & Technology Co., Ltd.). Lung tissue sections were sequentially immersed in xylene, absolute ethanol, 95% ethanol, and distilled water for dewaxing. The staining process followed the protocol of hematoxylin for 5 min, distilled water for 5 min, 75% ethanol for 2 s, 95% ethanol for 2 min, and eosin for 3 min. The sections were then sequentially immersed in 95% ethanol, absolute ethanol, and xylene for transparency, and finally, mounted in neutral resin for preservation.

During the immunohistochemistry experiment, lung tissue sections were sequentially immersed in xylene, absolute ethanol, 95% ethanol, and distilled water for dewaxing. An antigen retrieval solution was prepared, and the sections were subjected to boiling for 2 min. After rinsing with PBS and removing the excess liquid, the sections were incubated with an endogenous peroxidase blocking agent at room temperature for 10 min. Following another PBS rinse and the removal of excess liquid, the sections were incubated with a non-specific blocking agent for 10 min. After removing the liquid from the sections, they were incubated with the primary antibody overnight at 4 °C. The primary antibody was then collected, and the sections were washed with PBST, with the excess liquid removed, before being incubated with a biotinylated secondary antibody at room temperature for 10 min. After another PBST rinse and the removal of excess liquid, the sections were incubated with streptavidin-horseradish peroxidase at room temperature for 10 min. Following a PBST rinse, freshly prepared DAB was used to incubate the sections for 3–5 min before discarding it. The sections were then rinsed with distilled water and counterstained with hematoxylin for approximately 10 s. Finally, the sections were sequentially immersed in 95% ethanol, absolute ethanol, and xylene for transparency, and mounted in neutral resin for preservation.

### 2.5. Hydroxyproline Assay

The right lung of the mouse was isolated and placed at the bottom of a 5 mL glass vial, which was then dried in a 120 °C oven for 18 h. Subsequently, 2 mL of 6 M hydrochloric acid was added, and the vial was tightly sealed to prevent the evaporation of the hydrochloric acid, followed by heating in a 120 °C oven for another 18 h. After that, 2 mL of 6 M sodium hydroxide solution was added, the mixture was filtered through a 0.45 µm filter, and the pH was adjusted to between 6.8 and 7.4 before diluting to a final volume of 10 mL.

A standard curve was established using hydroxyproline standard solutions ranging from 0.625 µg/mL to 10 µg/mL. For the prepared standard solutions, 400 µL was transferred to an Eppendorf tube, while 50 µL of the sample to be measured was added to another Eppendorf tube, followed by the addition of 350 µL of ultrapure water. Subsequently, 200 µL of chloramine T was added to each tube, and the mixture was vortexed and allowed to sit at room temperature for 20 min. After that, 200 µL of HClO_4_ was added to each tube, which was vortexed and left at room temperature for an additional 5 min. Next, 200 µL of P-DMAB was added, followed by vortexing and an incubation in a 50 °C water bath for 20 min to produce a red compound. After cooling, 200 µL was taken from each Eppendorf tube and transferred to a 96-well plate to measure the absorbance at 577 nm.

### 2.6. Cell Culture, Transfection, and Treatment

NIH-3T3 cells and MLg cells were purchased from the ATCC cell bank. Both cell types were cultured in complete RPMI 1640 medium supplemented with 10% FBS (10099-141, Gibco, GrandIsland, NY, USA).

Cells in the culture plates were transfected when they reached 80–90% confluence. Ten microliters of siRNA (10 µM) was added to 250 µL of Opti-MEM (1897175, Gibco) and gently mixed; then, 5 µL of transfection reagent (40802ES, Yeasen Biotechnology (Shanghai) Co., Ltd., Shanghai, China) was added to another 250 µL of Opti-MEM and incubated at room temperature for 5 min. The two solutions were then combined and gently mixed, followed by a 20 min incubation at room temperature to allow the formation of RNA–lipid complexes. The prepared complexes were uniformly added to different regions of each well, and the culture plates were gently agitated. After 6 h, the medium was replaced with fresh culture medium or treatment with drugs was initiated. Two pairs of siRNAs targeting the FAP-α mRNA were selected for RNA interference in NIH-3T3 cells. The expression levels of FAP-α were detected using Western blot analysis, and the pair of siRNAs with the most effective interference was chosen for subsequent experiments, with the sequences being the sense strand 5′-GAACCAGGAGCAUGUAGAA-3′ and the antisense strand 5′-UUCUACAUGCUCCUGGUUC-3′.

### 2.7. Rapid and Sensitive Detection of Cell Proliferation and Cytotoxicity

A suspension of the passaged cells was obtained, counted, and diluted to a concentration of 7.5 × 10^4^ cells/mL. The suspension was mixed well and 100 µL was added to each well of a 96-well plate. After drug treatment of the cells for the expected duration, 10 µL of Cell Counting Kit-8 (CCK-8) reagent (C0005, Topscience Co., Ltd., Rizhao, China) was added to each well of the 96-well plate, and the mixture was thoroughly mixed to avoid bubble formation. The 96-well plate was then incubated in a cell culture incubator for 1 to 4 h, followed by shaking for 1 min. The absorbance at 450 nm was measured using a microplate reader.

### 2.8. Wound Healing Assay

Primary lung fibroblasts were seeded in plates. Before inflicting the wound, the cells were fully confluent. A scratch was made in the center of the culture well using a sterile 200 μL micropipette tip. The wounds were observed using an inverted optical microscope, and multiple images were obtained at areas flanking the intersections of the wound and the marker lines after the scratch at 0, 12, and 24 h. Images were obtained for analysis using Image-Pro Plus software 6.0.

### 2.9. Quantitative Real-Time PCR

For cell samples, after drug treatment, the original medium in the cell culture plate was discarded, and the cells were washed with PBS and aspirated. Then, 1 mL of Trizol (10606ES60, Yeasen Biotechnology (Shanghai) Co., Ltd.) was added to each well, and the cells were transferred to Eppendorf tubes by pipetting. For tissue samples, lung tissue was minced in Eppendorf tubes, and 0.4 mL of Trizol was added, followed by multiple rounds of homogenization and ultrasonic disruption on ice, with Trizol brought up to a final volume of 1 mL. Chloroform was added to the Trizol in the sample tubes, mixed by vortexing for 30 s, allowed to sit at room temperature for 3 min, and then centrifuged at 12,000 rpm for 15 min at 4 °C. At this stage, the solution separated into three layers. The upper aqueous phase, approximately 390 µL, was carefully transferred to a new Eppendorf tube, and an equal volume of isopropanol was added, mixed gently, and allowed to sit at room temperature for 10 min. The mixture was then centrifuged at 12,000 rpm for 10 min at 4 °C, and the supernatant was discarded. To the white pellet, 1 mL of 75% ethanol was added, followed by centrifugation at 7000 rpm for 5 min at 4 °C. This step was repeated once more, with the supernatant being carefully aspirated using a white pipette tip during the second round. The pellet was dried for 20 min to allow for complete evaporation of the ethanol, and then 10–20 µL of RNase-free H_2_O was added to dissolve the RNA. After sitting at room temperature for 10 min to ensure complete dissolution, the RNA concentration was measured.

Reverse transcription was performed according to the instructions of the FastKing gDNA Dispelling RT SuperMix (KR118, TIANGEN, Beijing, China) kit, setting up a 20 µL reaction system. After establishing the system, the reaction was carried out at 42 °C for 15 min followed by 95 °C for 3 min.

A 20 µL reaction system was set up according to the instructions of the Hieff^®^ qPCR SYBR Green Master Mix (No Rox) (11201ES03, Yeasen Biotechnology (Shanghai) Co., Ltd.). The reaction program was as follows: 5 min of pre-denaturation at 95 °C, followed by 10 s of denaturation at 95 °C, 20 s of annealing at 60 °C, and 20 s of extension at 72 °C, for a total of 45 cycles. The primer sequences used in the experiment are listed in Table 1.

### 2.10. Western Blotting

First, prepare an ice box and perform the subsequent operations on ice. For the cell culture plates after drug treatment, discard the original culture medium and rinse with PBS, then aspirate. To each well of a six-well plate, add 70 µL of RIPA lysis buffer (1% NaF, 1% PMSF), scrape the cells repeatedly with a pre-chilled cell scraper, and transfer the lysate to an Eppendorf tube. The cells are then subjected to multiple rounds of ultrasonic disruption. The samples are centrifuged at 12,000 rpm for 10 min at 4 °C. After centrifugation, the supernatant is carefully collected into a new Eppendorf tube, and the protein concentration is determined using the BCA assay. Based on the volume of the collected supernatant, add 5× loading buffer and mix thoroughly by pipetting. Heat the protein samples at 100 °C for 7 min.

Mouse lung tissues were dissected and separated, then snap-frozen in liquid nitrogen, and subsequently stored at −80 °C. A portion of the lung tissue was minced and then mixed with RIPA lysis buffer (1% NaF, 1% PMSF), followed by multiple rounds of tissue homogenization and ultrasonic disruption. The samples were centrifuged at 12,000 rpm for 10 min at 4 °C, with subsequent operations performed in accordance with the protocol for total protein extraction from cells.

In a 96-well plate, standard curve wells and sample wells were designed. The procedure involved sequentially adding PBS diluent, standard protein, or sample protein, and BCA working solution. The plate was incubated at 37 °C for 30 min, and the absorbance was measured at 562 nm.

Based on the protein concentration, the loading amount was calculated, with 30 µg of sample added to each well, flanked by pre-stained proteins on both sides. Appropriate-sized PVDF membranes were cut and activated in methanol. On the transfer membrane apparatus, layers of sponge, filter paper, gel, PVDF membrane, filter paper, and sponge were placed in order. After clamping, the assembly was inserted into the transfer tank. Transfer was conducted at 100 V for 150 min, ensuring that the transfer tank was placed in an ice-water bath. A TBST solution containing 5% non-fat milk was prepared and used to cover the PVDF membrane for a 1 h incubation.

Primary antibodies against collagen I (AF7001, Affinity), α-SMA (AF1032, Affinity), Fibronectin (AF0738, Affinity), P-Smad2 (Ser467) (AF3449, Affinity), Smad2 (AF6449, Affinity), P-Smad3 (Ser423 + Ser425) (AF8315, Affinity), Smad3 (AF6362, Affinity), SQSTM1/p62 (AF5384, Affinity), Beclin 1 (AF5128, Affinity), cleaved Caspase 3 (Asp175) (AF7022, Affinity), LC3A/B (D3U4C) (12741T, Cell Signaling Technology), cleaved Caspase 3 (Asp175) (AF7022, Affinity), Caspase 3/p17/p19 (19677-1-AP, Proteintech), Caspase 9 (AF6348, Affinity), cleaved Caspase 9 (Asp353) (AF5240, Affinity), P-mTOR (Ser2448) (5536S, Cell Signaling Technology), mTOR (2983S, Cell Signaling Technology), P-S6 Ribosomal Protein (Ser235/236) (4858S, Cell Signaling Technology), S6 Ribosomal Protein (AF6354, Affinity), P-ULK1 (Ser757) (14202T, Cell Signaling Technology), ULK1 (8054T, Cell Signaling Technology), P-p70 S6 Kinase (Thr389) (9234T, Cell Signaling Technology), p70 S6 Kinase (AF6226, Affinity), and FAP (66562, Cell Signaling Technology) were prepared according to the instruction manual (typically at a dilution of 1:1000) and dissolved in 5% non-fat milk. The PVDF membrane was incubated with the primary antibody overnight at 4 °C. After removal from 4 °C, the primary antibody was allowed to warm to room temperature before being collected, and the membrane was washed with TBST. Secondary antibodies were selected based on the origin of the primary antibody and incubated with the membrane at room temperature for 2 h. Following the collection of the secondary antibody, the membrane was washed with TBST, and detection reagent was added. Imaging was performed using a chemiluminescence apparatus (Beijing Sage Creation Science Co., Ltd., Beijing, China), and gray scale analysis was conducted using Image J 1.8.0 software.

### 2.11. Flow Cytometry

After drug exposure, the culture plates were collected, and the procedure was carried out according to the instructions of the apoptosis detection kit (CA1020, Beijing Solarbio Science & Technology Co., Ltd.). For adherent cells, the cell culture medium was collected into a centrifuge tube, and trypsin (without EDTA) was added to digest the cells. The previously collected culture medium was injected into the corresponding wells, and the cells were blown and collected into the centrifuge tube. The samples were centrifuged at 1000 rpm for 5 min, and the supernatant was carefully aspirated to avoid disturbing the cells. The cells were resuspended in 1 mL of pre-cooled PBS at 4 °C and centrifuged again to remove the supernatant. The cells were then resuspended in binding buffer, adjusting the concentration to 1–5 × 10^6^ cells/mL. A volume of 100 µL of the cell suspension was transferred to a 5 mL flow cytometry tube, and 5 µL of Annexin V/FITC was added and mixed well, followed by a 5 min incubation at room temperature, avoiding light exposure. Then, 5 µL of propidium iodide (PI) was added, followed by the addition of 400 µL of PBS. The wavelengths for FITC and PI were selected, and the flow cytometry analysis was conducted immediately.

### 2.12. Molecular Docking

The crystal structure of the target protein FAP-α (ID: 1Z68, Organism: Homo sapiens) was downloaded from the Protein Data Bank website, and the molecular structure of oleuropein (Compound CID: 5281544) was downloaded from the PubChem website. Molecular docking was performed using AutoDock Tools 4.2.6 software, following the sequence of ligand preparation, receptor preparation, grid file setup, and docking file configuration (using the Genetic Algorithm). The docking results were obtained and analyzed accordingly. The specific interactions between oleuropein and the target were examined using PyMOL 3.1.0 software.

### 2.13. Statistical Analysis

The research data are presented as means ± SDs and analyzed using GraphPad Prism 9 software. Differences between the control group and the model group, as well as between the drug treatment group and the model group, were evaluated using one-way ANOVA, with *p* < 0.05 considered statistically significant. Cell experiments were repeated three times, and the sample size for animal experiments was no less than five subjects.

## 3. Results

### 3.1. Oleuropein Can Alleviate Bleomycin-Induced Pulmonary Fibrosis in Mice

To investigate the antifibrotic effects of oleuropein in vivo, bleomycin was administered via intratracheal injection on day 0 to induce pulmonary fibrosis in mice. From days 7 to 13, different doses of oleuropein or the positive control drug nintedanib were given to the mice via gavage. The structural formula of oleuropein is shown in Figure 1A. On day 14, the mice were sacrificed for evaluation, as illustrated in Figure 1A.

The progression of idiopathic pulmonary fibrosis (IPF) is associated with alterations in lung function, which we evaluated in mice by measuring forced vital capacity (FVC). The results are shown in Figure 1B. The BLM model caused a significant decrease in FVC. Oleuropein treatment resulted in a dose-dependent increase in FVC, with the improvements in the Ole 50 mg/kg and Ole 100 mg/kg groups comparable to those in the Nin 60 mg/kg group. Hydroxyproline, a unique amino acid in collagen, can be quantitatively measured in lung tissue to indirectly reflect collagen deposition during the fibrotic process. As illustrated in Figure 1C, the hydroxyproline content was significantly increased in the BLM group compared to the NaCl group. Oleuropein treatment reduced the hydroxyproline content, with the Ole 50 mg/kg group showing a significant decrease and outperforming the Nin 60 mg/kg group. Mouse left lung sections were prepared for hematoxylin–eosin staining, as shown in Figure 1D. The bleomycin group exhibited thickening of the alveolar septa and extensive fibrotic lesions, while treatment with oleuropein effectively alleviated the extent of pulmonary fibrosis. Based on the H&E staining results, the statistical data on the percentage of lung fibrosis area is presented in Figure 1E. Compared to the BLM model group, the percentages of lung fibrotic areas in the Ole 50 mg/kg and Ole 100 mg/kg groups were significantly reduced, with both groups exhibiting greater improvements than the Nin 60 mg/kg group. The Masson staining method yielded similar results (Figure 1F,G).

In summary, oleuropein treatment significantly reduced hydroxyproline levels in lung tissue and the percentage of the lung fibrotic area, effectively improved the weight loss and pulmonary function parameters in mice, and alleviated the extent of pulmonary fibrosis (Table 2).

### 3.2. Oleuropein Inhibits TGF-β1-Induced Fibroblast Migration and Activation in Vitro

To further evaluate the effects of oleuropein on fibroblasts involved in the process of pulmonary fibrosis, we conducted in vitro studies using mouse embryonic fibroblast NIH-3T3 cells and mouse lung fibroblast MLg cells. Firstly, we assessed the impact of oleuropein at concentrations ranging from 2.5 µM to 320 µM on the viability of NIH-3T3 and MLg cells using the CCK-8 assay. The results are presented in Figure 2A, indicating that oleuropein at concentrations from 2.5 µM to 160 µM had no significant effect on the viability of NIH-3T3 and MLg cells, while 320 µM oleuropein began to exhibit cytotoxic effects. Based on literature reports of in vitro studies with oleuropein and the preliminary experimental results, we selected concentrations of 5 µM, 10 µM, and 20 µM for subsequent in vitro studies. We employed a scratch assay to evaluate the effect of oleuropein on TGF-β1-induced fibroblast migration, with the results for NIH-3T3 and MLg cells corresponding to Figure 2B,C. The results demonstrate that TGF-β1 induction significantly enhanced the migratory capacity of NIH-3T3 and MLg cells, facilitating rapid healing of the scratch towards the center compared to the control group. Oleuropein was found to dose-dependently inhibit the migration of both NIH-3T3 and MLg cells.

In healing tissues, fibroblasts acquire a contractile phenotype and transform into myofibroblasts, characterized by the formation of stress fiber bundles and the re-expression of α-smooth muscle actin (α-SMA). Myofibroblasts participate in the repair response by secreting extracellular matrix (ECM) components, including collagen I and fibronectin, and are responsible for the contraction of healing wounds. However, prolonged or excessive activation of myofibroblasts can lead to fibrosis. After TGF-β1 induced the activation of NIH-3T3 and MLg fibroblasts, we assessed the effects of oleuropein on the activation marker α-SMA as well as the ECM components collagen I and fibronectin. In NIH-3T3 cells, TGF-β1 induction significantly increased the expression of all three markers compared to the control group, while oleuropein dose-dependently inhibited the expression of these markers (Figure 3A, B). Real-time quantitative PCR results (Figure 3C) corroborated these findings. Following TGF-β1 induction, the protein expression and transcription of α-SMA, collagen I, and fibronectin were elevated in MLg cells. Oleuropein also dose-dependently inhibited the transcript and protein expression levels of all three markers (Figure 3D–F), indicating that oleuropein effectively suppresses fibroblast activation and ECM deposition at both transcriptional and protein expression levels.

In summary, oleuropein inhibited TGF-β1-induced fibroblast migration, activation, and ECM deposition in vitro.

### 3.3. Oleuropein Inhibits the TGF-β1/Smad Pathway in Fibroblasts

The TGF-β/Smad pathway plays a crucial role in the activation of myofibroblasts and the progression of pulmonary fibrosis. We assessed the impact of oleuropein on the TGF-β/Smad pathway in NIH-3T3 and MLg cells (Figure 4A,B). The results indicated that oleuropein dose-dependently inhibited the phosphorylation of Smad2 and Smad3 without affecting their expression levels. Therefore, oleuropein dose-dependently suppresses the TGF-β1/Smad pathway in fibroblasts, thereby inhibiting fibroblast migration, activation, and ECM deposition through these pathways.

### 3.4. Oleuropein Improves the Reduced Autophagy and Apoptotic Resistance in Fibroblasts Induced by TGF-β1 in Vitro

During the process of pulmonary fibrosis, fibroblasts and pulmonary epithelial cells exhibit autophagy dysfunction. Beclin1, LC3, and p62 are three key proteins involved in the autophagy process, and are used to assess autophagy levels. In the initiation phase of autophagy, Beclin1 forms a trimeric complex with VPS34 and Atg14, recruiting autophagy-related proteins to mediate the onset of autophagy. The LC3 precursor is first processed into soluble LC3-I, which is subsequently conjugated with phosphatidylethanolamine (PE) to form lipid-soluble LC3-II-PE, contributing to the formation of the autophagosome membrane. The levels of LC3-II or the ratio of LC3-II/LC3-I are positively correlated with the number of autophagosomes. P62, located in the cytoplasm, binds to ubiquitinated proteins and subsequently forms a complex with LC3-II, which is ultimately degraded within lysosomes. In autophagy, the degradation of p62 in the autophagic vesicles by lysosomes leads to a decrease in p62 levels.

Chloroquine (CQ) serves as an autophagy inhibitor. Figure 5A demonstrates that oleuropein inhibited the chloroquine-induced elevation in p62 levels in MLg cells, indicating that oleuropein enhances autophagy in fibroblasts. Subsequently, TGF-β1 was used to induce a decrease in autophagy in NIH-3T3 and MLg fibroblasts, and the effects of oleuropein on the three autophagy markers were assessed, as shown in Figure 5B,C. Oleuropein dose-dependently reduced the levels of p62 while increasing the Beclin1 levels and the ratio of LC3-II/LC3-I. Therefore, oleuropein effectively ameliorates the autophagy dysfunction in fibroblasts associated with pulmonary fibrosis.

During the process of fibrosis, alveolar epithelial cells are prone to apoptosis, while fibroblasts/myofibroblasts exhibit resistance to apoptosis. The caspase protein family plays a critical role in the apoptotic process, with cleaved Caspase 9 and cleaved-Caspase 3 serving as activated forms of Caspase 9 and Caspase 3, respectively; their increased levels indicate the occurrence of apoptosis. TGF-β1 was used to induce apoptosis resistance in NIH-3T3 and MLg fibroblasts, and the effects of oleuropein on this process were assessed. Oleuropein dose-dependently increased the levels of cleaved Caspase 9 and cleaved Caspase 3, thereby improving the TGF-β1-induced resistance to apoptosis in fibroblasts (Figure 6A,B).

Apoptosis was assessed using the Annexin V-FITC/PI apoptosis detection kit in conjunction with flow cytometry. Early apoptotic cells were identified by positive Annexin V-FITC staining and were located in the lower right quadrant of the scatter plot. The results indicated that TGF-β1 reduced the incidence of early apoptosis in fibroblasts, while treatment with oleuropein increased the levels of early apoptosis (Figure 6C). In summary, oleuropein effectively ameliorates autophagy dysfunction and apoptosis resistance in fibroblasts associated with pulmonary fibrosis.

### 3.5. Oleuropein Inhibits the TGF-β1/mTOR Signaling Pathway in Fibroblasts

mTOR activity may be dysregulated in fibroblasts from individuals with idiopathic pulmonary fibrosis (IPF), leading to a fibroblast phenotype characterized by increased proliferation and resistance to apoptosis through alterations in autophagy activity. mTOR complex 1 (mTORC1) responds to signals such as nutrients and growth factors by phosphorylating downstream substrates, thus inhibiting autophagy. The activation of the mTOR pathway is accompanied by the phosphorylation of downstream targets, including UNC-51-like kinase (ULK1), p70 S6 kinase (p70S6k), and S6 ribosomal protein (S6RP). In NIH-3T3 and MLg fibroblasts, the mTOR pathway was activated by TGF-β1, and the effects of oleuropein on this pathway were assessed. The results, shown in Figure 7A,B, indicate that oleuropein dose-dependently decreased the levels of mTOR, ULK1, p70 S6k, and S6RP phosphorylation in fibroblasts, thus inhibiting the TGF-β1/mTOR pathway. Therefore, it is inferred that oleuropein improves autophagy dysfunction and apoptosis resistance in fibroblasts by inhibiting the TGF-β1/mTOR pathway.

### 3.6. Oleuropein Effectively Alleviates BLM-Induced Pulmonary Fibrosis In Vivo via the TGF-β1/Smad and TGF-β1/mTOR Signaling Pathways

After inducing pulmonary fibrosis in mice using bleomycin, oleuropein treatment was administered, and following the treatment, the lung tissues of the mice were collected for an in vivo pharmacological evaluation. Western blot analysis of the lung tissues (Figure 8A) showed that oleuropein dose-dependently inhibited the increased expression of the fibrosis-related fibroblast activation markers collagen I and α-SMA while also suppressing the elevated levels of the autophagy marker p62. The inhibitory effects on these markers were comparable to those of the positive control drug, nintedanib, indicating that oleuropein effectively suppresses both fibrosis-induced fibroblast activation and autophagy dysfunction in vivo. Real-time quantitative PCR and immunohistochemistry analyses of lung tissue (Figure 8B–D) also demonstrated that oleuropein dose-dependently inhibited the transcription and expression of the fibroblast activation markers collagen I and α-SMA.

The effects of oleuropein on the TGF-β1/Smad pathway and the TGF-β1/mTOR pathway were further validated using the aforementioned mouse lung tissues. Western blot analysis of lung tissues (Figure 8E) demonstrated that oleuropein dose-dependently inhibited the phosphorylation of Smad2 and Smad3, indicating that oleuropein effectively suppresses the TGF-β1/Smad pathway in vivo. Additionally, oleuropein also dose-dependently inhibited the phosphorylation of mTOR and S6RP (Figure 8F), confirming that oleuropein effectively suppresses the TGF-β1/mTOR pathway in vivo.

In summary, in vivo pharmacological studies have demonstrated that oleuropein inhibits fibrosis-induced fibroblast activation and enhances autophagy levels. Additionally, oleuropein suppresses the TGF-β1/Smad pathway and the TGF-β1/mTOR pathway.

### 3.7. Oleuropein Targets the FAP-α Protein to Alleviate Pulmonary Fibrosis

The above experiments indicate that oleuropein inhibits the activation of fibroblasts via the TGF-β1/Smad pathway while enhancing fibroblast autophagy and apoptosis through the TGF-β1/mTOR pathway, thereby exerting its anti-pulmonary fibrosis effects. Subsequently, we aimed to identify the specific targets of oleuropein. Target prediction and molecular docking analyses revealed that fibroblast activation protein-α (FAP-α) may potentially interact with the compound. The molecular docking results demonstrated a binding affinity coefficient of −7.19 between FAP-α and oleuropein, indicating a strong binding interaction. In FAP-α, the residues LEU-206, ALA-207, PHE-350, THR-354, ILE-367, ASN-399, and ILE-400 form hydrogen bonds with oleuropein, as shown in Figure 9A.

siRNA was employed to interfere with the expression of FAP-α in NIH-3T3 fibroblasts to verify whether oleuropein exerts its anti-fibrotic effects through the FAP-α target. First, the efficacy of siRNA interference was assessed using Western blot analysis. As shown in Figure 9B, siRNA-2 demonstrated significant interference and was selected for subsequent experiments. In NIH-3T3 cells, the effects of oleuropein treatment or FAP-α interference on the fibroblast activation marker α-SMA and the TGF-β1/Smad pathway were evaluated. As illustrated in Figure 9C, D, the inhibitory effects of oleuropein treatment, FAP-α interference, or the combination of oleuropein and FAP-α interference on α-SMA expression and the level of Smad3 phosphorylation were comparable to those observed with TGF-β1 treatment alone. Additionally, the effects of oleuropein treatment or FAP-α interference on the expression of the autophagy marker p62, the apoptosis marker cleaved Caspase 9, and the TGF-β1/mTOR pathway were examined in NIH-3T3 cells. As displayed in Figure 9E, F, the increase in p62 levels and the inhibitory effects on cleaved Caspase 9, mTOR, and S6RP phosphorylation observed with oleuropein treatment, FAP-α interference, or their combination were similar to those seen with TGF-β1 treatment alone. These results indicate that oleuropein targets FAP-α to inhibit fibroblast activation via the TGF-β1/Smad pathway, while also enhancing autophagy and apoptosis in fibroblasts through FAP-α’s action on the TGF-β1/mTOR pathway, thereby exerting its anti-pulmonary fibrosis effects.

In summary, molecular docking confirmed strong binding affinity between FAP-α and oleuropein. In vitro knockdown of FAP-α indicated that oleuropein inhibits fibroblast activation through the TGF-β1/Smad pathway by suppressing FAP-α, and enhances fibroblast autophagy and apoptosis via its action on the TGF-β1/mTOR pathway through FAP-α (Figure 10).

## 4. Discussion

The lung interstitium is the supportive connective tissue and spaces surrounding the bronchi or blood vessels, alveolar septa, and under the visceral pleura, comprising the alveolar septa, lobular septa, and the peripheral tissues of bronchi or vessels. Idiopathic pulmonary fibrosis (IPF) is a chronic, irreversible, and deadly lung disease characterized by scarring and thickening of pulmonary interstitial tissue, leading to respiratory difficulty and ultimately respiratory failure. The histological features of IPF are defined by a usual interstitial pneumonia (UIP) pattern, which includes honeycombing in the lower lobes and subpleural regions, as well as foci of myofibroblasts located within more densely fibrotic areas, with an increase in proliferative type II alveolar epithelial cells and a decrease in type I alveolar epithelial cells [14,15]. Currently, only pirfenidone and nintedanib have been approved for the treatment of IPF, and while they demonstrate some efficacy in slowing the decline in lung function over a year, their limited effectiveness and serious side effects indicate a continued urgent need for new drugs in the IPF field.

Oleuropein is widely distributed in the stems, leaves, and fruits of Oleaceae plants, such as *Olea europaea* L. var, and exhibits various pharmacological activities, including anti-cancer, antioxidant, anti-inflammatory, cardiovascular-protective, neuroprotective, antiviral, skin-protective, and anti-aging effects [16,17]. Studies have shown that oleuropein exhibits anti-fibrotic effects on animal models of liver fibrosis arising from non-alcoholic steatohepatitis and cardiac fibrosis due to diabetes, although its impact on pulmonary fibrosis has been less frequently reported [9,10,11,12,13]. In respiratory diseases, oleuropein has been shown to suppress lung inflammation in animal models of asthma and chronic obstructive pulmonary disease (COPD) and provides protective effects on acute lung injury through its anti-inflammatory and antioxidant properties [11,12,13]. Early screenings in our study demonstrated that oleuropein significantly inhibited the expression of α-SMA in fibroblasts induced by TGF-β1. In a bleomycin-induced mouse model of pulmonary fibrosis, oleuropein treatment dose-dependently reduced hydroxyproline levels in lung tissue and the proportion of lung fibrotic area, improved lung tissue pathological features, and significantly alleviated weight loss and pulmonary function impairment in mice.

This study assessed the activation of fibroblasts to myofibroblasts using three markers: α-SMA, collagen I, and fibronectin. Various types of collagen and fibronectin are important extracellular matrix (ECM) proteins expressed by myofibroblasts. A prominent feature of myofibroblasts is the new expression of α-SMA in stress fibers, which serves as the molecular basis for their high contractile activity. During normal wound healing, myofibroblasts arise through various pathways, primarily through the differentiation of fibroblasts, and are regulated by factors such as TGF-β, WNT, fibronectin, and tissue stiffness. TGF-β is the principal growth factor guiding the formation of myofibroblasts, as it directly induces the production of extracellular matrix (ECM) and the expression of α-SMA. During the fibrotic process, TGF-β regulates multiple downstream signaling pathways, including the Smad3, PI3K/AKT, and p38 MAPK pathways, which promote the transdifferentiation of fibroblasts into myofibroblasts [18,19]. Notably, the overexpression of Smad3 enhances the production of α-SMA and ECM proteins by fibroblasts.

In vitro studies in this research demonstrated that oleuropein dose-dependently inhibited the TGF-β1-induced migration of fibroblasts and significantly suppressed the upregulation of the activation markers α-SMA, collagen I, and fibronectin at both the transcript and protein expression levels. Additionally, oleuropein dose-dependently inhibited the phosphorylation of Smad2 and Smad3 in TGF-β1-induced fibroblasts, indicating its inhibitory effect on the TGF-β1/Smad signaling pathway. Thus, oleuropein inhibits fibroblast migration, activation, and ECM deposition by suppressing the TGF-β1/Smad pathway.

Research also indicates that TGF-β can reduce autophagy by inhibiting the expression of several autophagy-related proteins (ATGs), including ATG 5 and ATG 7, as well as p62, in normal pulmonary fibroblasts [20]. The inhibition of autophagy through the knockout of LC3B and ATG 5 was found to increase the expression of myofibroblast markers in fibroblasts from patients with idiopathic pulmonary fibrosis (IPF) [21]. The fibroblasts from IPF patients naturally exhibit a sustained reduction in autophagy, which is associated with a propensity for fibrosis [22]. Fibronectin, a core component of the extracellular matrix (ECM), primarily regulates cellular activities through interactions with integrin receptors on the cell surface. In fibroblasts, the binding of fibronectin to integrins can mediate fibroblast survival, whereas the absence of this binding can lead to autophagy. Studies have shown that fibronectin can inhibit autophagy by activating the AKT/mTOR signaling pathway [23].

The in vitro studies conducted in this project demonstrated that oleuropein dose-dependently reduced p62 levels and increased the Beclin1 level and LC3-II/LC3-I ratio, indicating that oleuropein effectively improved TGF-β1-induced autophagy dysfunction in fibroblasts. Furthermore, oleuropein also dose-dependently elevated the protein levels of cleaved Caspase 9 and cleaved Caspase 3, as well as increased the proportion of early apoptotic cells in the Annexin V-FITC/PI apoptosis assay, suggesting that oleuropein alleviated TGF-β1-induced apoptotic resistance in fibroblasts. Additionally, oleuropein dose-dependently decreased the phosphorylation levels of mTOR, ULK1, p70S6K, and S6RP in fibroblasts, thereby inhibiting the TGF-β1/mTOR pathway. Consequently, oleuropein improves autophagic suppression and apoptotic resistance in fibroblasts during pulmonary fibrosis through the modulation of the TGF-β1/mTOR pathway.

FAP-α is a protein characterized by an α/β hydrolase domain and a β-spiral structure, playing a role in the fibrotic process. The levels of FAP-α regulate collagen integrity and ECM processing [24,25]. Typically absent in normal tissues, FAP-α is selectively expressed during tissue remodeling and repair, with its inhibition linked to cellular reprogramming and functional restoration [26]. The expression of FAP-α is associated with TGF-β signaling and inflammatory pathways, although the mechanisms of its inhibition remain unclear and may involve estrogen signaling and specific microRNAs [27,28,29]. Studies also indicate that the transcription of the FAP-α gene is regulated by various transcription factors, and its interaction with the chaperone protein BAG6/BAT3 may influence its degradation [30]. In pathological tissues, elevated expression of FAP-α is typically detrimental; thus, inhibiting FAP-α is considered beneficial for the treatment of inflammatory diseases such as metabolic syndrome, cancer, and fibrosis [24,25].

Molecular docking studies have demonstrated that FAP-α exhibits a strong binding affinity for oleuropein, with a binding energy of −7.19. In vitro knockout of FAP-α using siRNAs indicates that oleuropein inhibits the activation of fibroblasts by targeting the TGF-β1/Smad pathway through the suppression of FAP-α. Additionally, oleuropein enhances fibroblast autophagy and apoptosis via the TGF-β1/mTOR pathway, which is also mediated by FAP-α. FAP-α activity exhibits pro-fibrogenic functions and acts as a regulator of cell apoptosis, adhesion, and migration, independent of its enzymatic activity; however, there are currently no reports detailing how FAP-α influences the regulation of the TGF-β1/Smad and TGF-β1/mTOR pathways. It remains unclear whether FAP-α directly regulates the proteins within these pathways or exerts an indirect effect, and the specific regulatory mechanisms warrant further investigation [31,32,33].

In this study, we explored the therapeutic effects of oleuropein on a bleomycin-induced mouse model of pulmonary fibrosis. The results showed that oleuropein effectively alleviated weight loss and lung pathological changes caused by fibrosis, significantly reducing the fibrotic area and the tissue hydroxyproline content, while also improving lung function. These findings provide important experimental evidence for the potential of oleuropein as a therapeutic agent for pulmonary fibrosis. Further pharmacological studies, both in vitro and in vivo, indicated that oleuropein inhibits fibroblast migration, activation, and extracellular matrix deposition by suppressing the TGF-β1/Smad pathway. Additionally, oleuropein improves the suppression of autophagy and apoptotic resistance in fibroblasts by inhibiting the TGF-β1/mTOR pathway. This mechanism of action was further validated through molecular docking and in vitro studies, demonstrating a high affinity between oleuropein and FAP-α, with a binding energy of −7.19. The research suggests that oleuropein, by inhibiting FAP-α, not only acts on the TGF-β1/Smad pathway to suppress fibroblast activation but also enhances autophagy and apoptosis in fibroblasts through FAP-α.

A notable constraint of this investigation lies in its dependence on the bleomycin-induced pulmonary fibrosis model. Although this approach provides a robust platform for assessing therapeutic potential, it may not entirely recapitulate the intricate pathophysiology and phenotypic diversity observed in IPF or other clinical variants of the disease. Moreover, the relatively short treatment and observation windows could impede a comprehensive evaluation of the long-term therapeutic outcomes and potential adverse effects. This study’s focus on predefined endpoints may also have overlooked certain mechanistic or secondary biological responses associated with the interventions. Finally, while the sample size was adequate for initial hypothesis testing, it may limit the broader applicability and statistical power of the findings.

## 5. Conclusions

In summary, our findings demonstrate that oleuropein alleviates bleomycin-induced pulmonary fibrosis in mice by directly binding to FAP-α, thereby interfering with the TGF-β/Smad pathway and regulating fibroblast autophagy and apoptosis. Oleuropein has the potential to become a novel therapeutic agent for pulmonary fibrosis, and its mechanism of action provides a theoretical basis for future clinical applications.

## Figures and Tables

**Figure 1 biomolecules-15-01211-f001:**
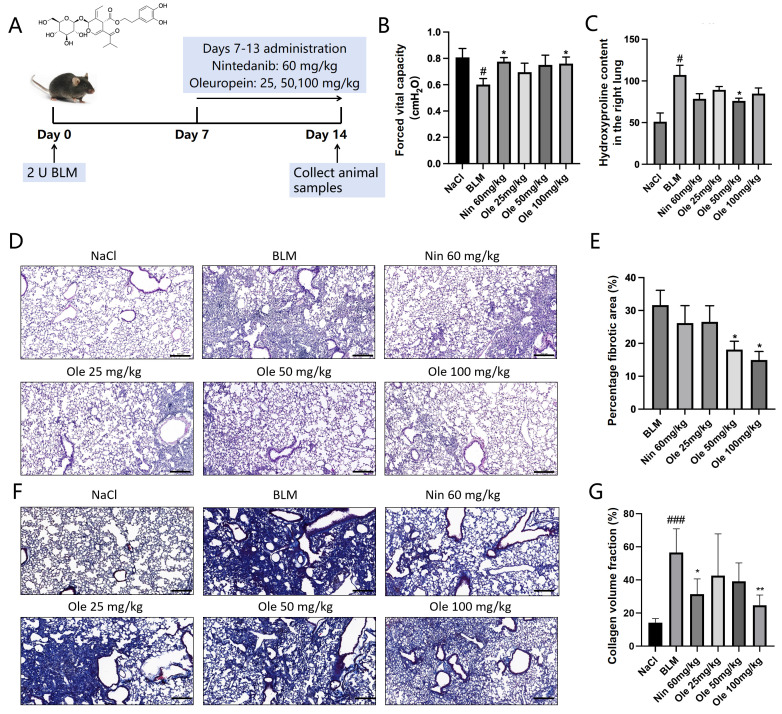
Effects of oleuropein on bleomycin-induced pulmonary fibrosis in mice in vivo. (**A**) Experimental protocol for the efficacy evaluation of the BLM-induced pulmonary fibrosis model in mice. (**B**) The results of the pulmonary function test for forced vital capacity. (**C**) Hydroxyproline content measurement in lung tissue. (**D**) H&E staining results (scale bar: 200 μm). (**E**) Percentage of lung fibrotic area. The data are presented as Mean ± SD (one-way ANOVA with Tukey’s post hoc multiple comparison tests). (**F**) Masson’s staining results (scale bar: 200 μm). (**G**) Quantification of the collagen density in images of Masson’s trichrome-stained lung tissues. n = 5. # indicates differences between the NaCl group and the BLM group, with # *p* < 0.05, and ### *p* < 0.001. * indicates differences between the BLM group and the oleuropein or nintedanib treatment groups, with * *p* < 0.05, ** *p* < 0.01.

**Figure 2 biomolecules-15-01211-f002:**
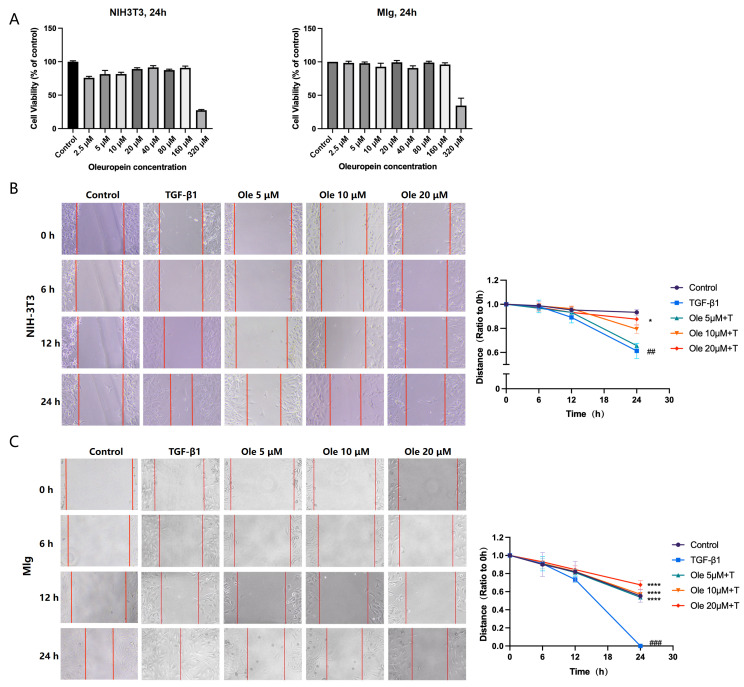
Effects of oleuropein on fibroblast viability and migration. (**A**) NIH-3T3 and MLg cells were exposed to oleuropein at concentrations ranging from 2.5 μM to 320 μM for 24 h, and cell viability was assessed using the CCK-8 assay. NIH-3T3 (**B**) and MLg (**C**) cells were exposed to oleuropein and/or TGF-β1 (5 ng/mL) for 24 h, with scratch wound healing assessed at 0 h, 6 h, 12 h, and 24 h. The data are presented as Mean ± SD (one-way ANOVA with Tukey’s post hoc multiple comparison tests), n = 3. # indicates differences between the control group and the TGF-β1 group, with ## *p* < 0.01, and ### *p* < 0.001. * indicates differences between the TGF-β1 group and the oleuropein treatment groups, with * *p* < 0.05, and **** *p* < 0.0001.

**Figure 3 biomolecules-15-01211-f003:**
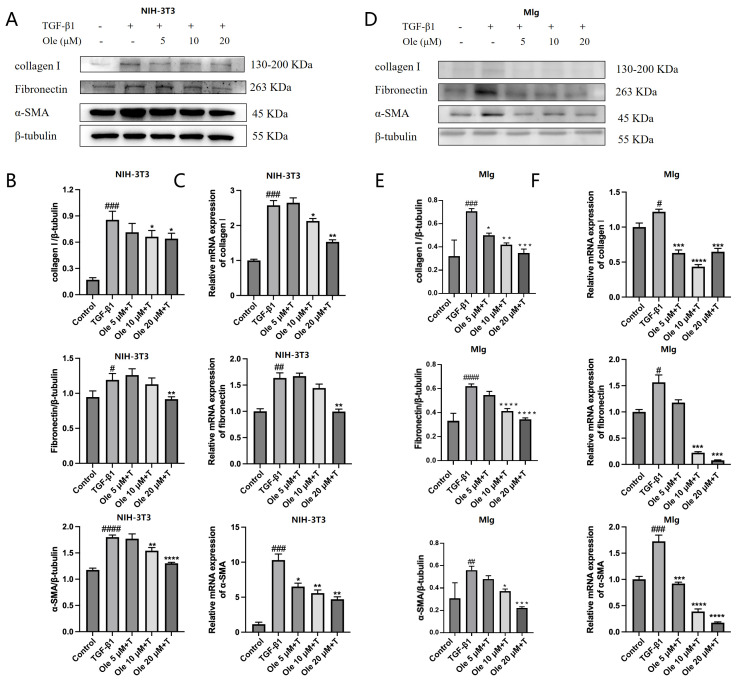
Effects of oleuropein on fibroblast activation and ECM deposition at the protein expression level. NIH-3T3 (**A**,**B**) and MLg (**D**,**E**) cells were exposed to oleuropein and/or TGF-β1 (5 ng/mL) for 24 h, after which samples were collected for Western blot analysis to assess the protein expression levels of collagen I, fibronectin, and α-SMA (original images can be found in Appendix A). NIH-3T3 (**C**) and MLg (**F**) cells were similarly treated, and samples were collected for real-time quantitative PCR to evaluate the RNA levels of collagen I, fibronectin, and α-SMA. The data are presented as Mean ± SD (one-way ANOVA with Tukey’s post hoc multiple comparison tests), n = 3. # indicates differences between the control group and the TGF-β1 group, with # *p* < 0.05, ## *p* < 0.01, ### *p* < 0.001, and #### *p* < 0.0001. * indicates differences between the TGF-β1 group and the oleuropein treatment groups, with * *p* < 0.05, ** *p* < 0.01, *** *p* < 0.001, and **** *p* < 0.0001.

**Figure 4 biomolecules-15-01211-f004:**
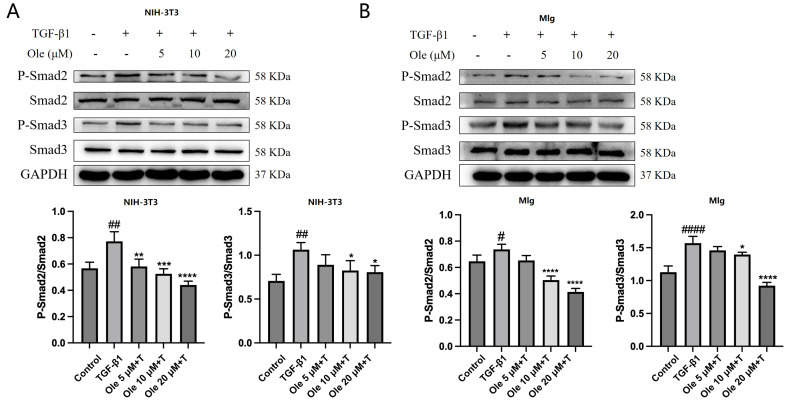
Effects of oleuropein on the TGF-β1/Smad pathway in fibroblasts. NIH-3T3 (**A**) and MLg (**B**) cells were exposed to oleuropein for 12 h and then treated with TGF-β1 (5 ng/mL) for 0.5 h before samples were collected for Western blot analysis to evaluate the protein expression levels of P-Smad2, Smad2, P-Smad3, Smad3 (original images can be found in Appendix A). The data are presented as Mean ± SD (one-way ANOVA with Tukey’s post hoc multiple comparison tests), n = 3. # indicates differences between the control group and the TGF-β1 group, with # *p* < 0.05, ## *p* < 0.01, and #### *p* < 0.0001. * indicates differences between the TGF-β1 group and the oleuropein treatment groups, with * *p* < 0.05, ** *p* < 0.01, *** *p* < 0.001, and **** *p* < 0.0001.

**Figure 5 biomolecules-15-01211-f005:**
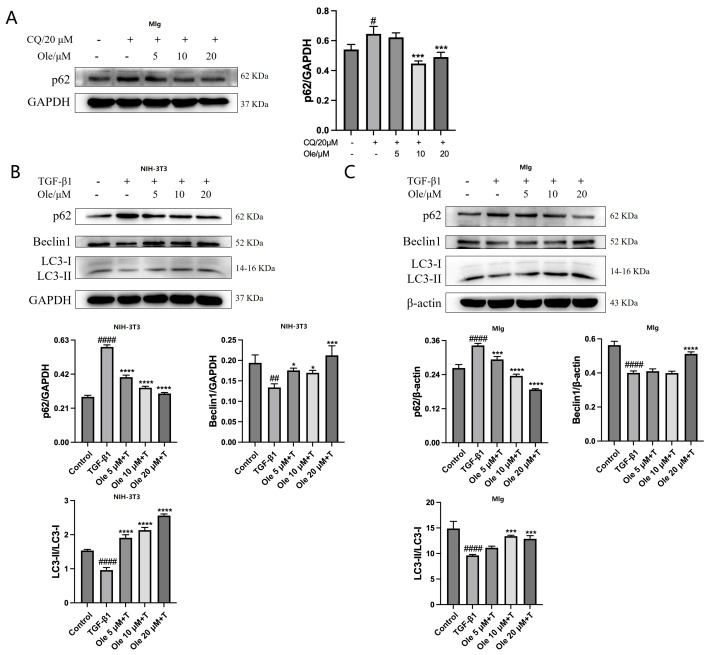
Oleuropein improves the reduction of autophagy in fibroblasts induced by TGF-β1 in vitro. (**A**) MLg cells were exposed to 20 μM chloroquine and/or oleuropein for 24 h, after which samples were collected for Western blot analysis to assess the protein expression levels of p62 (original images can be found in Appendix A). The results are presented as Mean ± SD, n = 3. # indicates differences between the control group and the CQ group, with # *p* < 0.05. * indicates differences between the CQ group and the oleuropein treatment group, with * *p* < 0.05, and *** *p* < 0.001. NIH-3T3 (**B**) and MLg (**C**) cells were exposed to oleuropein and/or TGF-β1 (5 ng/mL) for 24 h, followed by sample collection for Western blot analysis to evaluate the protein expression levels of Beclin1, LC3, and p62 (original images can be found in Appendix A). The results are presented as Mean ± SD, n = 3. # indicates differences between the control group and the TGF-β1 group, with ## *p* < 0.01, and #### *p* < 0.0001. * indicates differences between the TGF-β1 group and the oleuropein treatment group, with * *p* < 0.05, *** *p* < 0.001, and **** *p* < 0.0001.

**Figure 6 biomolecules-15-01211-f006:**
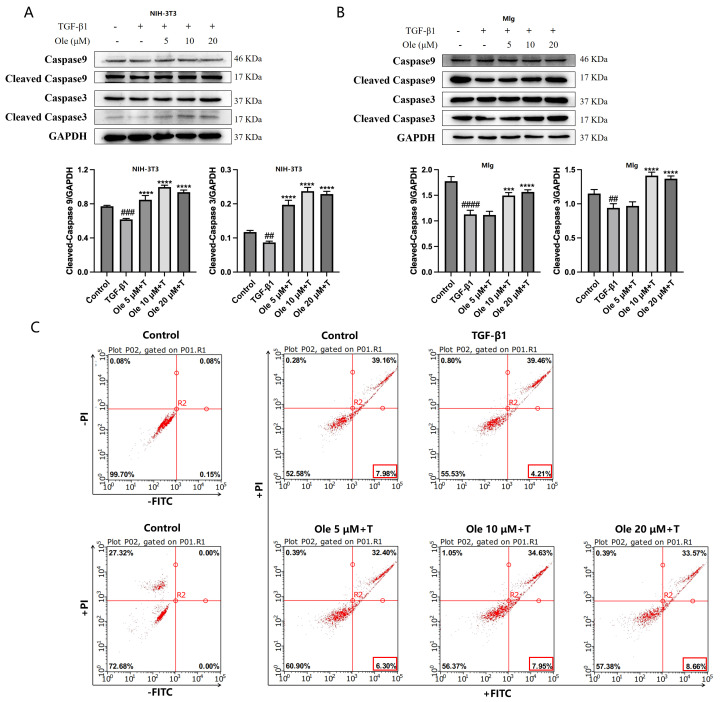
Oleuropein improves the resistance to apoptosis in fibroblasts induced by TGF-β1 in vitro. NIH-3T3 (**A**) and MLg (**B**) cells were exposed to oleuropein and/or TGF-β1 (5 ng/mL) for 24 h, after which samples were collected for Western blot analysis to evaluate the protein levels of cleaved Caspase 9, Caspase 9, cleaved Caspase 3, and Caspase 3 (original images can be found in Appendix A). The results are presented as Mean ± SD, n = 3, where # indicates differences between the control group and the TGF-β1 group, with ## *p* < 0.01, ### *p* < 0.001 and #### *p* < 0.0001, and * indicates differences between the TGF-β1 group and the oleuropein treatment group, with *** *p* < 0.001, and **** *p* < 0.0001. (**C**) MLg cells were exposed to oleuropein and/or TGF-β1 (5 ng/mL) for 24 h, after which cells were digested and collected for flow cytometry analysis using the Annexin V-FITC/PI apoptosis detection method.

**Figure 7 biomolecules-15-01211-f007:**
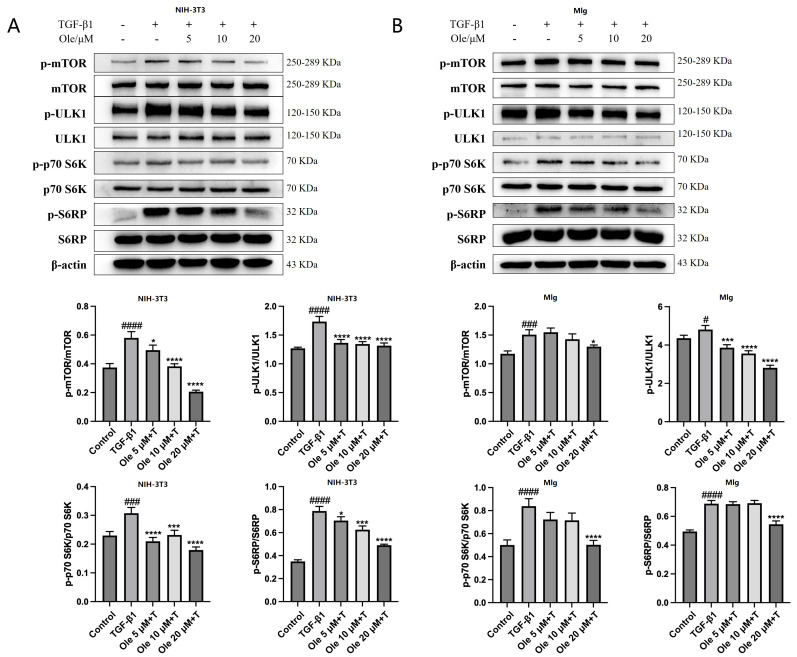
Oleuropein inhibits the TGF-β1/mTOR pathway in fibroblasts. NIH-3T3 (**A**) and MLg (**B**) cells were exposed to oleuropein and/or TGF-β1 (5 ng/mL) for 12 h, after which samples were collected for Western blot analysis to assess the protein levels of mTOR, p-mTOR, ULK1, p-ULK1, p70 S6 kinase, p-p70 S6k, S6 ribosomal protein, and p-S6RP (original images can be found in Appendix A). The results are presented as Mean ± SD, n = 3, where # indicates differences between the control group and the TGF-β1 group, with # *p* < 0.05, ### *p* < 0.001, and #### *p* < 0.0001, and * indicates differences between the TGF-β1 group and the oleuropein treatment group, with * *p* < 0.05, *** *p* < 0.001, and **** *p* < 0.0001.

**Figure 8 biomolecules-15-01211-f008:**
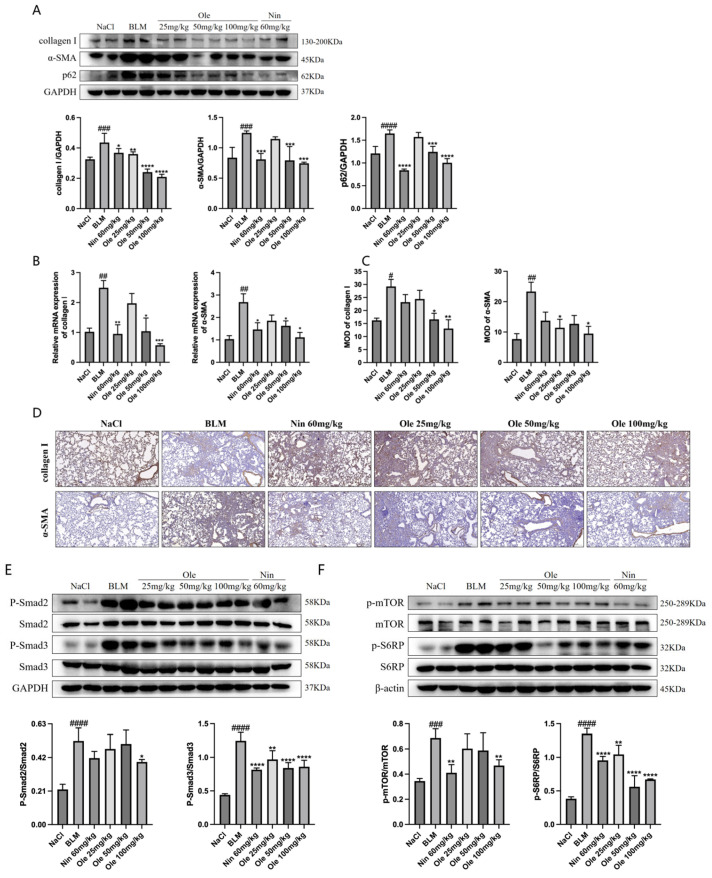
Oleuropein inhibits fibroblast activation and improves autophagy dysfunction in vivo. Lung tissues from mice with bleomycin-induced pulmonary fibrosis treated with oleuropein were collected for in vivo pharmacological assessments. (**A**) Western blot analysis was conducted on lung tissues to measure the protein expression levels of collagen I, α-SMA, and p62 (original images can be found in Appendix A). (**B**) Real-time quantitative PCR was performed on lung tissues to evaluate the transcript levels of collagen I and α-SMA. (**C**,**D**) Immunohistochemical staining of lung tissue sections was carried out to assess the protein levels of collagen I and α-SMA, with the semi-quantitative analysis of the staining results shown in (**C**). (**E**) Western blot analysis was conducted on lung tissues to detect the protein expression levels of P-Smad2, Smad2, P-Smad3, and Smad3 (original images can be found in Appendix A). (**F**) Western blot analysis was performed on lung tissues to measure the protein expression levels of mTOR, p-mTOR, S6RP, and p-S6RP (original images can be found in Appendix A). The results are presented as Mean ± SD, n = 5, where # indicates differences between the NaCl group and the BLM group, with # *p* < 0.05, ## *p* < 0.01, ### *p* < 0.001, and #### *p* < 0.0001, and * indicates differences between the BLM group and the treatment group, with * *p* < 0.05, ** *p* < 0.01, *** *p* < 0.001, and *****p* < 0.0001.

**Figure 9 biomolecules-15-01211-f009:**
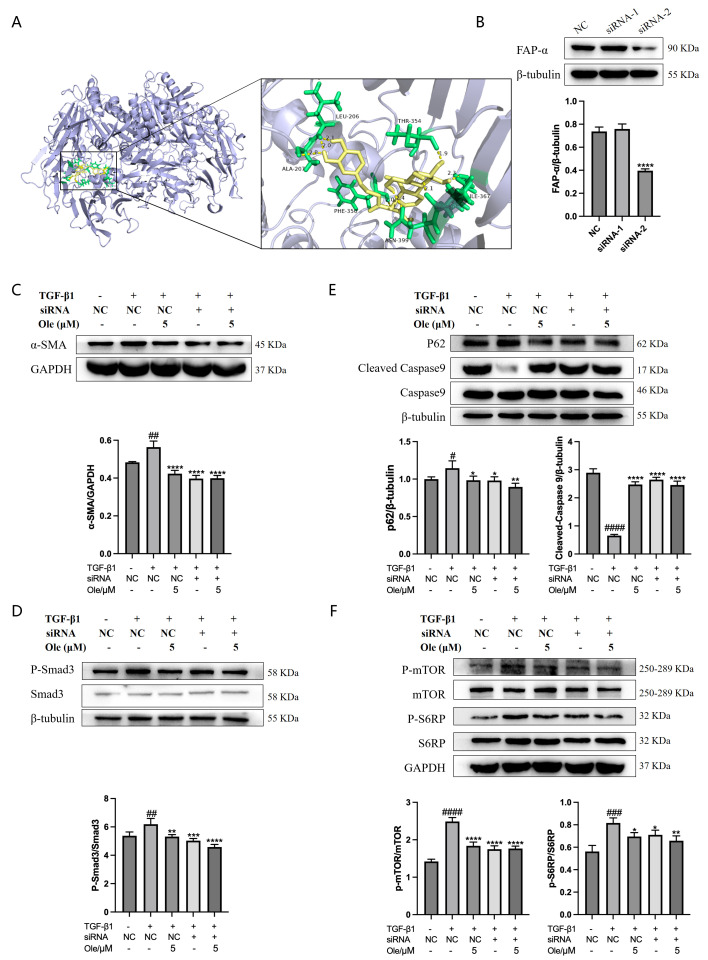
Oleuropein alleviates pulmonary fibrosis by binding to the FAP-α target. (**A**) Molecular docking results show FAP-α in blue and oleuropein in yellow, with hydrogen bonds formed between oleuropein and the amino acid residues of FAP-α indicated by green residues involved in hydrogen bonding. (**B**) NIH-3T3 cells were treated with FAP-α siRNA, and samples were collected for Western blot analysis to detect the expression levels of FAP-α and β-tubulin (original images can be found in Appendix A). (**C**) Following FAP-α siRNA treatment, NIH-3T3 cells were exposed to oleuropein and/or TGF-β1 (5 ng/mL) for 24 h, after which samples were collected for Western blot analysis to measure α-SMA expression (original images can be found in Appendix A). (**D**) After FAP-α siRNA treatment, NIH-3T3 cells were exposed to oleuropein for 12 h, followed by TGF-β1 (5 ng/mL) incubation for 0.5 h, and samples were collected for Western blot analysis to detect the expression levels of P-Smad3 and Smad3 (original images can be found in Appendix A). (**E**) Following FAP-α siRNA treatment, NIH-3T3 cells were exposed to oleuropein and/or TGF-β1 (5 ng/mL) for 24 h, and samples were collected for Western blot analysis to assess the expression levels of p62, cleaved Caspase 9, and Caspase 9 (original images can be found in Appendix A). (**F**) After FAP-α siRNA treatment, NIH-3T3 cells were exposed to oleuropein for 12 h, followed by TGF-β1 (5 ng/mL) incubation for 0.5 h, and samples were collected for Western blot analysis to examine the expression levels of mTOR, p-mTOR, S6RP, and p-S6RP (original images can be found in Appendix A). The results are presented as Mean ± SD, n = 3, where # indicates differences between the control and TGF-β1 groups, with # *p* < 0.05, ## *p* < 0.01, ### *p* < 0.001, and #### *p* < 0.0001, and * represents differences between the TGF-β1 group and the oleuropein and/or siRNA groups, with * *p* < 0.05, ** *p* < 0.01, *** *p* < 0.001, and **** *p* < 0.0001.

**Figure 10 biomolecules-15-01211-f010:**
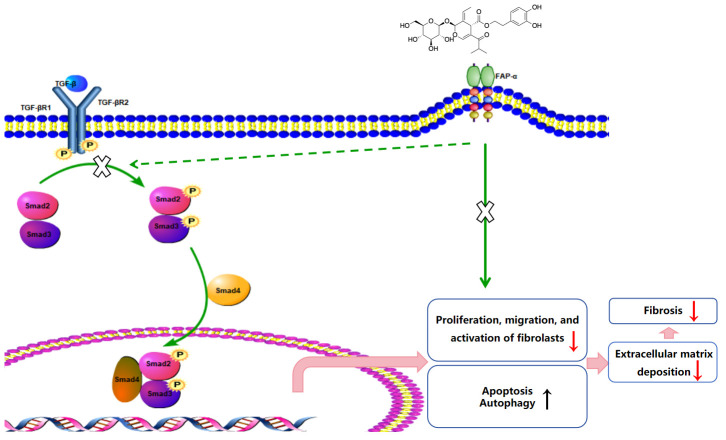
The mechanism by which oleuropein alleviates BLM-induced pulmonary fibrosis in mice.

**Table 1 biomolecules-15-01211-t001:** Primer sequences.

Gene (Mouse)	Forward Primer Sequence	Reverse Primer Sequence
	(5′-3′)	(5′-3′)
Fibronectin	GTGTAGCACAACTTCCAATTACGAA	GGAATTTCCGCCTCGAGTCT
Collagen I	CCAAGAAGACATCCCTGAAGTCA	TGCACGTCATCGCACACA
α-SMA	GCTGGTGATGATGCTCCCA	GCCCATTCCAACCATTACTCC
GAPDH	AGGTCGGTGTGAACGGATTTG	TGTAGACCATGTAGTTGAGGTCA

**Table 2 biomolecules-15-01211-t002:** Summary of the key results.

	NaCl	BLM	Nin (60 mg/kg)	Ole (25 mg/kg)	Ole (50 mg/kg)	Ole (100 mg/kg)	F	*p*
**FVC (mmH_2_O** **)**	0.81 ± 0.10	0.60 ± 0.10 ^#^	0.78 ± 0.06 *	0.70 ± 0.12	0.75 ± 0.13	0.76 ± 0.10 *	1.304	0.0318
**Hyp (ug/right lung)**	51.10 ± 14.82	107.19 ± 23.35 ^#^	78.61 ± 12.04	89.31 ± 8.10	76.01 ± 6.65 *	84.83 ± 13.40	5.173	0.0027
**Collagen Volume Fraction (%)**	14.26 ± 2.20	56.57 ± 12.82 ^###^	31.40 ± 8.29 *	42.54 ± 22.65	39.10 ± 10.05	24.78 ± 5.48 **	5.912	0.0011

# indicates differences between the NaCl group and the BLM group, with # *p* < 0.05, ### *p* < 0.001, and * indicates differences between the BLM group and the treatment group, with * *p* < 0.05, ** *p* < 0.01.

## Data Availability

The original contributions presented in this study are included in the article/Appendix A. Further inquiries can be directed to the corresponding author.

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
