# Peer review of "Oleuropein Ameliorates Bleomycin-Induced Pulmonary Fibrosis in Mice by Targeting TGF-β1 Signaling Pathway"

_biomolecules, 2025, doi:10.3390/biom15091211_

Round 1
Reviewer 1 Report
Comments and Suggestions for Authors
This study explores the therapeutic potential of oleuropein, a natural compound derived from olive leaves, in the treatment of idiopathic pulmonary fibrosis (IPF). Using a bleomycin-induced mouse model, the authors show that oleuropein significantly reduces pulmonary fibrosis, improves lung function, and decreases collagen deposition. In vitro studies with NIH-3T3 and MLg fibroblasts demonstrate that oleuropein inhibits TGF-beta1-induced fibroblast activation, migration, and extracellular matrix production. Molecular docking and siRNA experiments suggest fibroblast activation protein-a (FAP-a) as a putative target. Collectively, the findings provide preclinical evidence supporting oleuropein as a potential anti-fibrotic agent via modulation of fibroblast function and TGF-beta1 signaling pathways.
Minors:
- H&E staining is not appropriate for quantitative assessment of fibrosis. Masson's trichrome or picrosirius red (PSR) staining should be used to specifically visualize and quantify collagen fibers. H&E does not reliably distinguish fibrotic area, and thus cannot be used to generate accurate fibrosis area percentages.
- To accurately assess cell viability and morphology in the scratch assay, viability markers such as calcein-AM staining should be included. This would confirm that oleuropein's effects are not due to cytotoxicity and provide clearer evaluation of cell integrity.
- Figure 3A and B: There is inconsistency in the use of internal controls (beta-tubulin vs. GAPDH) across experiments. This should be standardized. Additionally, in Figure 3B, the GAPDH band in the first lane appears substantially weaker than in other lanes, raising concerns about loading consistency and the reliability of conclusions regarding TGF-beta1 induction.
Author Response
Comment 1: H&E staining is not appropriate for quantitative assessment of fibrosis. Masson's trichrome or picrosirius red (PSR) staining should be used to specifically visualize and quantify collagen fibers. H&E does not reliably distinguish fibrotic area, and thus cannot be used to generate accurate fibrosis area percentages.
Reply 1:We sincerely appreciate the reviewer’s insightful comment regarding the limitations of H&E staining for fibrosis quantification. We fully agree that H&E staining is not optimal for accurately assessing fibrotic areas, as it does not reliably distinguish collagen fibers. As suggested, we have now performed Masson’s trichrome staining to specifically visualize and quantify collagen deposition. The revised figures and results reflect these improvements, ensuring a more precise and reliable evaluation of fibrosis.
Thank you for this valuable suggestion, which has significantly strengthened the methodological rigor of our study.
Comment 2: To accurately assess cell viability and morphology in the scratch assay, viability markers such as calcein-AM staining should be included. This would confirm that oleuropein's effects are not due to cytotoxicity and provide clearer evaluation of cell integrity.
Reply 2:We appreciate the reviewer’s valuable suggestion regarding the assessment of cell viability in the scratch assay. As noted in our previous experiments, we conducted CCK-8 assays to evaluate the potential cytotoxicity of oleuropein at the concentrations used in this study. The results confirmed that the selected doses had no significant impact on cell viability, ensuring that the observed effects in the scratch assay were not attributable to cytotoxicity.
We thank the reviewer for this insightful suggestion, which has enhanced the robustness of our study.
Comment 3: Figure 3A and B: There is inconsistency in the use of internal controls (beta-tubulin vs. GAPDH) across experiments. This should be standardized. Additionally, in Figure 3B, the GAPDH band in the first lane appears substantially weaker than in other lanes, raising concerns about loading consistency and the reliability of conclusions regarding TGF-beta1 induction.
Reply 3:In response to the reviewer’s comment, we have now standardized our Western blot analyses by using beta-tubulin as the internal control in Figure 3D and 3E. This ensures uniformity in data presentation and interpretation.
The revised Figure 3D demonstrates uniform beta-tubulin expression across all lanes, reinforcing the reliability of our conclusions on TGF-β1 induction.
We thank the reviewer for these critical observations, which have significantly improved the robustness and reproducibility of our data. The revised figures and corresponding methodological details are now included in the updated manuscript.

Reviewer 2 Report
Comments and Suggestions for Authors
1) Please follow the ARRIVE checklist/reporting guidelines to structure this manuscript and attach a completed checklist as supplemental material.
2) There are too many experiments/aims within a single manuscript, making this a dense report and difficult to follow. Please consider showing a single table/figure (similar to a central graphical abstract) in which mention each aim, its corresponding experimental method, and its corresponding key finding.
3) Section 2.1: please provide references and a rationale to explain your choice of model- ie why choose a bleomycin PF mouse model instead of bleomycin induced PF in wistar rat or other models?
4) Section 2.2-2.11 please provide references and rationale for choices made in regard to the measurement techniques of different outcome and mediator variables.
5) Before the section 2.12 (statistical analysis) please add a separate section with the heading "experimental design" in which please describe the experimental design in detail- how many mice models were prepared with bleomycin- was PF confirmed and was the degree similar in all mice? was there randomization of mice to test drug vs control drug? why was an active comparator used and how was it selected? why a sham arm with no active comparator not included? was blinding implemented? at what stage and using what methods? was randomization of mice done? using which methods? (e.g. were sealed envelopes used?)
6) For the primary aim results (which are currently shown in Figure 1 graphs without accompanying numeric values)- please also add a table providing numeric estimates/values for key outcomes along with 95% CI and p-values, in tabular format.
7) At the end of discussion section, please add a para (10-15 lines) discussing the various assumptions made and limitations of this study (including both technical and methodological limitations such as explanations for how selection bias, misclassification bias, measurement bias, type I and type II error etc might occur and impact interpretations as well as impact on generalizability of findings)
Author Response
Comment 1: Please follow the ARRIVE checklist/reporting guidelines to structure this manuscript and attach a completed checklist as supplemental material.
Reply 1:We sincerely appreciate the reviewer’s suggestion to improve the transparency and reproducibility of our study by adhering to the ARRIVE (Animal Research: Reporting of In Vivo Experiments) guidelines.
We have carefully restructured the manuscript to fully comply with the ARRIVE guidelines, ensuring that all recommended sections (e.g., study design, ethical statements, sample size justification, randomization, blinding, and statistical methods) are now clearly and thoroughly addressed.
We are grateful for this valuable suggestion, which has significantly strengthened the rigor and reproducibility of our work. The revised manuscript and supplementary ARRIVE checklist now fully align with the highest standards of in vivo research reporting.
Comment 2: There are too many experiments/aims within a single manuscript, making this a dense report and difficult to follow. Please consider showing a single table/figure (similar to a central graphical abstract) in which mention each aim, its corresponding experimental method, and its corresponding key finding.
Reply 2:We sincerely appreciate the reviewer’s constructive feedback regarding the organization and clarity of our manuscript. We agree that consolidating the key experimental aims, methods, and findings into a single visual summary would significantly improve readability and help readers navigate the study more efficiently.
In response to this suggestion, we have now included a central graphical abstract in Figure10.
This new schematic provides a clear, high-level overview of the study’s design and major conclusions, ensuring that the logical flow of the work is immediately apparent. We have also streamlined the text in the Results and Discussion sections to better align with this structured presentation.
We are grateful for this insightful recommendation, which has greatly enhanced the accessibility and impact of our manuscript.
Comment 3: Section 2.1: please provide references and a rationale to explain your choice of model- ie why choose a bleomycin PF mouse model instead of bleomycin induced PF in wistar rat or other models?
Reply 3:Regarding the choice of the C57BL/6J strain, we acknowledge your concern about potential bias introduced by selecting a pro-inflammatory strain for our in vivo experiments. We understand that including a different strain, such as BALB/c, or considering an outbred strain may enhance the robustness of our findings by providing a broader perspective on the inflammatory response. Induction of pulmonary fibrotic lesions in C57BL/6J mice by BLM is a widely recognized and well-established model for studying IPF. Regarding the choice of mouse strains, studies indicate that C57BL/6J mice are a particularly suitable strain for this research[1-2]. Regarding the selection of mice and rats, we comprehensively considered the operability of the animals and referred to previous research articles. Ultimately, we chose mice as the main model organism. Thank you once again for your constructive criticism, which will significantly contribute to improving the quality and rigor of our research. We are committed to addressing these points thoroughly in the revision.
References:
[1]Matute-Bello G, Downey G, Moore BB, Groshong SD, Matthay MA, Slutsky AS, Kuebler WM; Acute Lung Injury in Animals Study Group. An official American Thoracic Society workshop report: features and measurements of experimental acute lung injury in animals. Am J Respir Cell Mol Biol. 2011 May;44(5):725-38.
[2]Tashiro J, Rubio GA, Limper AH, Williams K, Elliot SJ, Ninou I, Aidinis V, Tzouvelekis A, Glassberg MK. Exploring Animal Models That Resemble Idiopathic Pulmonary Fibrosis. Front Med (Lausanne). 2017 Jul 28;4:118.
Comment 4: Section 2.2-2.11 please provide references and rationale for choices made in regard to the measurement techniques of different outcome and mediator variables.
Reply 4:Thank you for your valuable feedback regarding the methodology section of our manuscript. We acknowledge that a detailed and transparent description of our experimental procedures is essential for reproducibility and for readers to fully evaluate our study.We appreciated your highlighting the need for major revisions, and we carefully reviewed and expanded the methodology section to provide comprehensive details.
Thank you again for your constructive suggestions, which will greatly improve the clarity and rigor of our work.
Comment 5: Before the section 2.12 (statistical analysis) please add a separate section with the heading "experimental design" in which please describe the experimental design in detail- how many mice models were prepared with bleomycin- was PF confirmed and was the degree similar in all mice? was there randomization of mice to test drug vs control drug? why was an active comparator used and how was it selected? why a sham arm with no active comparator not included? was blinding implemented? at what stage and using what methods? was randomization of mice done? using which methods? (e.g. were sealed envelopes used?)
Reply 5:Thank you for your valuable feedback regarding the experimental design section of our manuscript. We acknowledge that a detailed and transparent description of our experimental design is essential for reproducibility and for readers to fully evaluate our study.
Your specific comments were addressed point-by-point in our revised manuscript to ensure that all key aspects were thoroughly and clearly documented. We are committed to enhancing this section to meet the expected scientific standards.
Thank you again for your constructive suggestions, which will greatly improve the clarity and rigor of our work.
Comment 6: For the primary aim results (which are currently shown in Figure 1 graphs without accompanying numeric values)- please also add a table providing numeric estimates/values for key outcomes along with 95% CI and p-values, in tabular format.
Reply 6:Thank you for your valuable feedback regarding the clarity of the figures in our manuscript. We sincerely apologize for any difficulty this may have caused in interpreting our data.
We fully agree with your opinion and have added relevant content in the text, which made it easier for readers to understand the average value and the degree of data dispersion of each group. We hope that our revisions will make it easier for readers to understand our content.
We appreciate your constructive criticism, and we are committed to improving the visual quality of our figures in the revised manuscript. Thank you once again for your thoughtful suggestions.
Comment 7: At the end of discussion section, please add a para (10-15 lines) discussing the various assumptions made and limitations of this study (including both technical and methodological limitations such as explanations for how selection bias, misclassification bias, measurement bias, type I and type II error etc might occur and impact interpretations as well as impact on generalizability of findings)
Reply 7:Thank you for your insightful feedback regarding the inclusion of a section on the limitations of our study in the Discussion. We appreciate your suggestion, as a thoughtful reflection on the limitations is essential to contextualizing our findings and guiding future research.
In our revised manuscript, we dedicated a paragraph to discuss the limitations we encountered during our study. This included considerations related to experimental design, sample size, the generalizability of our findings, and any other relevant factors that may have impacted the interpretation of our results. By transparently addressing these limitations, we aimed to provide a more balanced perspective on our work and its implications.
Thank you once again for your valuable input, which will enhance the depth and rigor of our Discussion section. We are committed to incorporating this feedback into our revision.

Round 2
Reviewer 2 Report
Comments and Suggestions for Authors
Thank you for submitting a revised manuscript. I have no additional comments